# SpotLight Proteomics—A IgG-Enrichment Phenotype Profiling Approach with Clinical Implications

**DOI:** 10.3390/ijms20092157

**Published:** 2019-05-01

**Authors:** Susanna L. Lundström, Tina Heyder, Emil Wiklundh, Bo Zhang, Anders Eklund, Johan Grunewald, Roman A. Zubarev

**Affiliations:** 1Division of Physiological Chemistry I, Department of Medical Biochemistry and Biophysics, Karolinska Institutet, 17177 Stockholm, Sweden; Bo.Zhang@ki.se; 2Respiratory Medicine Unit, Department of Medicine Solna & Centre for Molecular Medicine, Karolinska Institutet, 171 77 Stockholm, Sweden; tina.heyder@ki.se (T.H.); emil.wiklundh@ki.se (E.W.); Anders.Eklund@ki.se (A.E.); Johan.Grunewald@ki.se (J.G.)

**Keywords:** IgG, sarcoidosis, Löfgrens syndrome, biomarker

## Abstract

Sarcoidosis is a systemic interstitial lung disease of unknown aetiology. Less invasive diagnostics are needed to decipher disease pathology and to distinguish sub-phenotypes. Here we test if SpotLight proteomics, which combines de novo MS/MS sequencing of enriched IgG and co-extracted proteins with subsequent label-free quantification of new and known peptides, can differentiate controls and sarcoidosis phenotypes (Löfgrens and non-Löfgrens syndrome, LS and nonLS). Intra-individually matched IgG enriched from serum and bronchial lavage fluid (BALF) from controls (*n* = 12), LS (*n* = 11) and nonLS (*n* = 12) were investigated. High-resolution mass-spectrometry SpotLight proteomics and uni- and multivariate-statistical analyses were used for data processing. Major differences were particularly observed in control-BALF versus sarcoidosis-BALF. However, interestingly, information obtained from BALF profiles was still present (but less prominent) in matched serum profiles. By using information from orthogonal partial least squares discriminant analysis (OPLS-DA) differentiating 1) sarcoidosis-BALF and control-BALF and 2) LS-BALF vs. nonLS-BALF, control-serum and sarcoidosis-serum (*p* = 0.0007) as well as LS-serum and nonLS-serum (*p* = 0.006) could be distinguished. Noteworthy, many factors prominent in identifying controls and patients were those associated with Fc-regulation, but also features from the IgG-Fab region and novel peptide variants. Differences between phenotypes were mostly IgG-specificity related. The results support the analytical utility of SpotLight proteomics which prospectively have potential to differentiate closely related phenotypes from a simple blood test.

## 1. Introduction

Sarcoidosis is a systemic inflammatory disease which is characterized by non-caseating granulomas that predominately involve lung and hilar lymph nodes. Clinically, sarcoidosis can be categorized as Löfgren syndrome (LS) and non-Löfgren syndrome (nonLS) [1]. LS has an acute disease onset, although usually a more favorable outcome compared to nonLS, particularly in patients expressing human leukocyte antigen (HLA) type HLA-DRB1*03 [2]. Although a substantial amount of research has been undertaken, the immunological aetiology behind sarcoidosis remains elusive. Sarcoidosis has conventionally been considered a T-cell mediated disease. Interestingly, studies have shown antibody recognition towards mycobacterial antigens in patients [3,4] and HLA derived peptides can stimulate T-cell response [5]. Furthermore, high frequencies of autoantigens have been reported in sarcoidosis bronchoalveolar lavage fluid (BALF) and serum [6]. By identifying exogenous peptides and autoantigens, these studies suggested a contribution of immunoglobulins (Ig) to disease pathogenesis.

We recently developed the SpotLight proteomics approach [7] that combines de novo MS/MS sequencing of enriched antibodies and co-extracted proteins with subsequent label-free quantification of new and known peptides. In a pilot study using this approach on differentiating two types of neurodegenerative disorders, the hidden proteome added almost as much information to patient stratification as the apparent proteome [7]. Intriguingly, many of the new peptide sequences were attributable to antibody variable regions, and are potential biomarkers indicative of disease aetiology. Here we test if the SpotLight approach is also applicable in differentiating two sarcoidosis phenotypes (i.e., LS vs. nonLS). Since sarcoidosis is a predominant chronic lung disorder, it would be expected that the IgG in closest proximity to the lung is more disease specific. However, since antibodies are circulating in the body, it is likely that features identified in lung-IgG can also be identified in blood-IgG. Given that diagnostic bronchoscopy with BAL can be unpleasant, easier sampling by a blood test would facilitate diagnostic procedures significantly. Thus, in order to test the hypothesis that the blood-IgG profiles prospectively can be used to differentiate sarcoidosis phenotypes in clinical samples, and thereby to identify and develop prognostic markers, we applied SpotLight proteomics on intra-individually matched samples from the lung (i.e., BALF) and blood (i.e., serum) of LS and nonLS sarcoidosis patients as well as of healthy controls. The SpotLight approach and study design are described in Figure 1. 

## 2. Results

### 2.1. Data Overview

In total, 825 IgG specific peptides (with conserved or novel sequence variations), 59 IgG-Fc-glycopeptides and 2369 peptides that either could be assigned to other known protein sequences or represented novel sequences were identified in at least 50% of the serum or BALF samples (Figure 2). Of this data, 1711 features could be detected both in serum and BALF, 1053 features were only found in serum, and 696 features were only found in BALF. The biggest differences between BALF and serum were found in non-IgG peptides, for which a greater portion 38% (*n* = 909) were found in serum compared to 24% (*n* = 580) in BALF. In contrast, of the IgGome and Fc-glycan peptides (that were detected in at least 50% of all individuals), the majority could be identified in both matrices (83%, *n* = 730). Principal component analysis (PCA) of the complete data set (3 components, R2 = 0.63, Q2 = 0.58) differentiated BALF and serum samples along the first component and BALF-controls and BALF-sarcoidosis patients along the second component (Figure 3A). When interrogating how the features assembled along the second component (Figure 3B), it became apparent that many of the IgG conserved, variable and novel peptide sequences are correlating with sarcoidosis. A complete list of all detected features, including subgroup averages, means and p-values are listed in Appendix A. Separate PCA analyses of the BALF and serum data sets are given in Appendix A. 

### 2.2. Differences Between Controls and Sarcoidosis Patients

From PCA analysis it is evident that the main differences between patients and controls were observed in samples taken in close proximity to the disease location (i.e. the BALF). 1072 features were found to be significantly different between the controls and sarcoidosis patients in the BALF. Of these, 724 remained significant following FDR- and 42 following BF correction (Figure 4A). For the serum samples, the results were more modest with 393 features identified as significantly different, and of which 16 remained significant following FDR-, and 5 following BF correction (Figure 4C). Noteworthy, in the BALF many of the FDR corrected features were from (*n* = 125) or showed sequence homology (*n* = 84) with IgG. Furthermore, out of the remaining FDR corrected features ~40% (*n* = 233) were from novel peptide sequences. It is likely that within this pool of peptides there might be hints on disease-specific proteoforms and CDR-chain variants. In contrast to BALF, no IgG related variable peptide chain sequences remained significant following correction in the serum. Instead, and as expected, the majority of significantly different features could be linked to inflammation. Noteworthy, when examining which features were consistently different in both matrixes two distinct trends were observed. Specifically, proteins/and peptides originating from the complement cascade were elevated in sarcoidosis patients as well as agalactosylated Fc-glycan peptides. For the BALF we could also identify an elevation in proteins/peptides in neutrophil activation/mediated immunity, cholesterol, lipid, and amyloid regulating pathways (Table 1, Figure 5). Furthermore, proteins/peptides involved in peptidase related activities were elevated in the sarcoidosis patients while the peptidase inhibitors were found in lower abundances (i.e., correlating with the healthy individuals). 

### 2.3. Differences Between LS and non-LS Patients

In contrast to the large differences between controls and sarcoidosis patients, the differences between the phenotypes were much weaker (Figure 4B,D, respectively). However, interestingly, the majority of factors differentiating nonLS and LS were observed within the portion of IgG related peptides and novel sequences (IgG+novel: 61%, IgG only: 40%) compared to factors differentiating controls and sarcoidosis patients (IgG+novel: 49%, IgG only: 19%). Thus, out of 78 significant factors (peptides) in BALF, 62 were from IgG and novel peptide sequences. For the serum, out of 102 significant factors, 57 were from IgG and novel peptides. Furthermore, out of the nine factors that were consistently different between nonLS and LS in both matrices, eight were from IgG, of which seven were from the variable regions. Noteworthy, in contrast to the big differences between patients and controls in BALF (Figure 4A) compared to serum (Figure 4C), more features were significantly different in serum (Figure 4D) than in BALF (Figure 4B) when comparing the phenotypes. However, considering that nothing remained significant following FDR correction, we knew that many of the identified factors should be interpreted with caution, as some are likely to be false positives. It is noteworthy to mention that the total profile information obtained from BALF (from which we obtained most differentiation according to the IgG profile) was still sufficient to discriminate the phenotypes in serum. For details on significant/non-significant features in the phenotypes, see Appendix A. 

### 2.4. Predicting Sarcoidosis Disease Status in Serum via Information Obtained from BALF

The main goal of this study was to test if information obtained by circulating polyclonal IgG (and co-eluting proteins) in BALF can be used to differentiate patient phenotypes and controls in serum. First, we tested this approach to differentiate serum-control and serum- sarcoidosis patients. The samples (that were treated as unknowns), were distinguished with a significance of *p* = 0.0007 (Figure 6A) using BALF-control and BALF-sarcoidosis information. In the next step, we tested the same approach to differentiate the serum-nonLS and serum-LS phenotypes using BALF information. Strikingly, even though the phenotype differences were much smaller than in the previous case, the model was still able to differentiate the phenotypes (*p* = 0.006), Figure 6B. Noteworthy, when testing the opposite approach and predicting the phenotypes in BALF by using the serum information, the phenotypes could still be distinguished, albeit with a lower significance (*p* = 0.03). 

## 3. Discussion

The primary aim of this study was to see if Melon Gel extracted IgG from serum could be used to differentiate phenotype specific sarcoidosis with the help of information obtained from Melon Gel extracted IgG from BALF. The utility of SpotLight proteomics has previously been demonstrated in differentiating two types of neurodegenerative diseases with 85%–95% accuracy [7]. Herein, the same approach was used but instead of working with serum samples only, we used BALF samples for multivariate model building and serum samples (treated as unknowns) for validation. Given that antibodies are circulating in the system from less accessible but disease-specific organs (i.e., lung), to the organs that can be easily sampled (i.e., blood), we hypothesized that even though the disease and phenotype-specific imprint or profile is likely weaker in blood, it should still be present. This hypothesis was validated on the serum enriched IgG profiles by using information from the IgG enriched lung profile (which in theory should have a stronger imprint). Strikingly, using this approach both controls and sarcoidosis patients as well as nonLS- versus LS-patients could be differentiated. We also tested the inverse, i.e., using the serum profile data to differentiate the BALF samples. As expected the results were less striking. This strengthens the hypothesis that most disease and phenotype-specific information should be extracted in close proximity to the disease. However, by gathering (and validating) this information, it can prospectively be used to differentiate the patients using their corresponding enriched IgG serum profile. Thus, opening up the possibility of a new approach for disease and phenotype characterization via blood test sampling.

When analyzing polyclonal enriched IgG, using a bottom-up proteomics approach with sensitive Orbitrap MS instrumentation, other proteins and peptides that co-elute with the IgG can be identified [7,8,9]. Of these proteins, we expect a significant portion to be either co-eluting via Fc- or most intriguing Fab- binding interactions (Figure 1C). In this study, we could find several immune effector processing proteins that interact with the IgG-Fc region to be significantly elevated in the sarcoidosis patients (Figure 5). This trend was also in line with what was observed for the IgG-Fc-glycosylation pattern which was containing a significantly lower distribution of galactosylated forms [10]. We could also identify an elevation in the sarcoidosis patients in proteins involved in neutrophil activation/mediated immunity-, as well as cholesterol-, lipid-, and amyloid-regulating pathways (Figure 5). Furthermore, many of the protein/peptide sequences that were found in higher abundance in sarcoidosis are involved in peptidase activity. In contrast, several peptidase inhibitors had higher abundance in the controls. This information is in line with several other studies [11,12,13,14,15]. 

Not surprisingly, both univariate and multivariate data only indicated few/weak differences between the sarcoidosis phenotypes and with no factors reaching significance following FDR correction. However, interestingly, a majority of the phenotype differentiating factors were from the IgG variable regions. This was most specifically the case for the phenotype-specific features identified in the BALF-profiles and for features that were found to be significant according to phenotype in both matrices. Similarly to the BALF and serum IgG-profile models built to distinguish controls and sarcoidosis, the BALF-phenotype information remained superior in distinguishing the serum-phenotypes compared to using the serum-phenotype information to distinguish the BALF-phenotypes. One reason for this is likely that we obtain a larger portion of true positives in BALF (which is closer to the disease source). Another reason for this could be that the IgG profile (which was more phenotype specific in the BALF), is also carried through the circulating blood system and thereby easier to target and extrapolate between the matrices. 

This pilot study was designed to test the hypothesis on a small number of patients and controls. The reproducibility of the results needs to be further validated on a larger cohort that includes both genders and with controls that are better age-matched to the patient group. However, importantly and as discussed above, most of the non-IgG detected features that were identified as sarcoidosis correlating have also been observed in previous studies [11,12,13,14,15]. Furthermore, the observed alteration in Fc-galactosylation status (which is known to be age dependent) remained significant following age correction [10]. 

In summary, we show the potential in using the serum profile of IgG and co-enriched proteins/peptides to target a disease or to characterize a disease phenotype. Within the pool of IgG and co-enriched proteins, we can identify both disease/phenotype specific features and features that will indicate the immune regulatory state of the patient. Considering that BALF sampling and other invasive sampling procedures can be daunting both for the patient and clinician, the idea of using easily accessible polyclonal IgG from a simple blood test is promising. Thus, we suggest that this approach prospectively can have huge potential and impact on disease/phenotype characterization. 

## 4. Material and Methods

### 4.1. Subject Information

Sarcoidosis patients and controls have previously been described in detail [10]. Briefly, patients with LS (*n* = 11) or non-LS (*n* = 12), and healthy controls (*n* = 12) underwent bronchoscopy with BAL as previously described [16]. Furthermore, one LS patient was sampled twice with one year between sampling dates. All subjects were non-smoking males of northern European descent. LS and non-LS were age-matched (LS: 42 ± 7 years, non-LS: 43 ± 9 years) while healthy subjects were significantly younger (27 ± 3 years). LS is defined via acute onset, usually, with fever, chest radiographic findings with bilateral hilar lymphadenopathy, sometimes with pulmonary infiltrates, and with erythema nodosum, or bilateral ankle arthritis. Sarcoidosis patients were diagnosed as defined by the World Association of Sarcoidosis and other Granulomatous Disorders (WASOG) [17]. Written informed consent was obtained from all participants. The study was approved by the ethical committee review board in Stockholm (2005/1031–31/2). Clinical characteristics of the subjects are described in Table 2.

### 4.2. Sample Preparations

Serum (40 µL/sample) and BALF (500 µL/sample) of sarcoidosis patients and controls were aliquoted in triplicates, prepared, and analyzed in a randomized manner as previously described [10]. Two replicates were analyzed by LC-MS/MS. The third (still available) set was confirmed to be consistent with the other two sets via pool analysis. Polyclonal IgGs were enriched from blood serum and from BALF using Melon Gel IgG Spin Purification Kit (Thermo Scientific) [7,10]. For serum IgG, ten µg of IgG/sample and for BALF the complete IgG samples were used for analysis. The protein S–S bonds were reduced, the cysteines alkylated, and the proteins cleaved with trypsin into peptides. Peptides were then desalted, dried down, and stored at −20 °C until LC-MS/MS analysis.

### 4.3. LC-MS/MS Analysis

Approximately 1 µg of digest/sample was analyzed using an EASY-nLC system connected to a Fusion Orbitrap mass-spectrometer (both Thermo Fisher Scientific) [10]. Briefly, reversed phase nano-LC-separation of the peptides was performed on a 50 cm long EASY spray column (PepMap, C18, 2 µm, 100 Å). The chromatographic separation was achieved using a solvent system containing (A) water with 2% acetonitrile and 0.1% formic acid and (B) acetonitrile with 2% water and 0.1% formic acid and with a flow rate of 300 nL/min. The mass spectrometer was operating in positive DDA mode. A survey mass spectrum was acquired in the *m*/*z* 200–2000 range with a nominal resolution of 120,000. Precursor ion selection was performed in “top speed” mode of 2 to 7 charged ions. Precursor ion selection for MS/MS, was performed for each precursor with both higher-energy collisional dissociation (HCD) and electron transfer dissociation (ETD). 

### 4.4. Protein and Peptide Identification and Quantification

A detailed description of the SpotLight approach including the de novo sequencing and quantitative analysis has been described in detail [7]. Briefly, all MS/MS spectra from the MG-extraction experiments were firstly searched against a human reference proteome with two missed cleavages as well as 10 ppm and 20 ppm mass tolerances for precursor and fragment peaks. Carbamidomethylation of cysteine was set as a fixed modification. Oxidation of methionine, deamidation of asparagine and glutamine, as well as acetylation of protein *N*-terminus were considered to be set as variable modifications. MS/MS spectra assigned to peptide sequences with <1% FDR were excluded. Remaining data underwent de novo sequencing with unassigned spectra submitted in a pair-wise (HCD-ETD) manner to pNovo + (v.1.3) [18]. Precursor mass range was set to 700–4000 Da with oxidized methionine considered to be an independent residue. Mass tolerance was set in MS/MS at 5 ppm for precursors and 15 ppm for fragments. Candidate sequences were filtered by the criteria of full backbone coverage. Three top-scoring peptide sequences were kept as candidates. Peptide candidates were homology-searched against the human UniProt protein database using BLASTp. The highest BLAST score match was reported as the final sequence. The UniProt human protein database and obtained de novo sequences were merged into the SpotLight database, on which a second database search was performed. For quantitation, raw data were processed through the DeMix-Q workflow [19,20], in which MS/MS spectra were matched against the SpotLight database. Protein abundances were calculated by averaging abundances of the three most abundant constituent peptides. Only proteins from which at least two unique peptides per protein would be quantified were used. Furthermore, for both protein and peptide data, only features found and quantified in at least 50% of either all serum- or all BALF samples were kept. The abundances of IgG peptides were re-normalized such that their total abundance in all samples was the same (100%). Non-IgG peptides, as well as proteins, were normalized separately in the same way. Assignment of complementarity determining regions (CDR) and framework regions (FR) were based on Uniprot information and by using the VBASE sequence directory (Tomlinson et al., MRC Centre for Protein Engineering, http://www2.mrc-lmb.cam.ac.uk/vbase/alignments2.php).

### 4.5. Fc-Glycopeptide Identification and Quantification

As previously described [9,21], 63 glycopeptide variants with glycans N-linked to tryptic-peptides EEQ**Y**NST**Y**R and TKPREEQ**Y**NST**Y**R (IgG_1_), EEQ**F**NST**F**R and TKPREEQ**F**NST**F**R (IgG_2_ or IgG_3_), as well as EEQ**F**NST**Y**R and TKPREEQ**F**NST**Y**R (IgG_4_) were screened. Glycopeptide ion abundances were integrated over the respective chromatographic peak of monoisotopic ions (<10 ppm) and within ±2 min retention-time interval. Glycan abundances were normalized to the total content of Fc-glycosylated IgG_1_ or IgG_2/(3)_ and IgG_4_ peptides. Glycan nomenclature is according to Royle et al. [22]. 

### 4.6. Statistical Analysis

Univariate analyses were performed using Student’s t-test (with equal or unequal variance depending on F-test). P-values were corrected according to Bonferroni (BF) and false discovery rate (FDR). Multivariate modeling using PCA and Orthogonal projections to latent structures discrimination analysis (OPLS-DA) was performed using SIMCA 15.0 following mean centering, log transformation, and UV scaling. OPLS-DA models were further validated using predicted scores t(tPS) of samples treated as unknowns. Pathway analysis was performed using STRING with a minimum required interaction score of 0.7. 

## Figures and Tables

**Figure 1 ijms-20-02157-f001:**
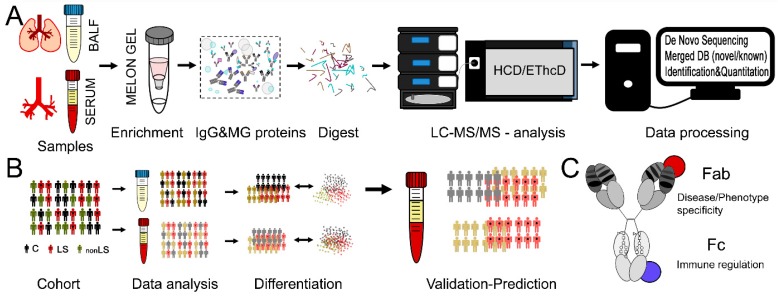
SpotLight approach and study design. (**A**) Samples from BALF and serum were IgG enriched, digested and subjected to LC-MS/MS analysis using HCD and EThcD fragmentation for de novo sequencing. New and known peptide sequences were merged into one database that was used for data identification and subsequent quantification. (**B**) Controls and sarcoidosis patients suffering from either nonLS or LS were selected. From each individual two intra-individually matched samples were obtained, one from lung (BALF) and one from blood (serum). Both sample types were subjected to differential multivariate data analysis. The results from BALF were then used to predict/differentiate the serum profiles that were treated as unknowns. (**C**) Different features measured via SpotLight proteomics. Fab sequence variation, potential Fab binding proteins and peptides, Fc-region variation, Fc-glycosylation pattern as well as immune regulatory Fc-binding proteins.

**Figure 2 ijms-20-02157-f002:**
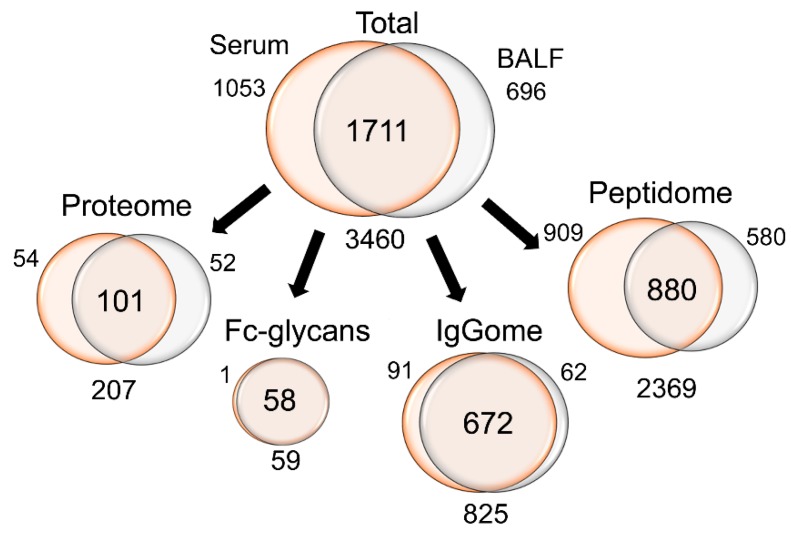
Data overview. Overview of the number of features that were acquired from the two sample types. Note that the identified features needed to be found in at least 50% of all serum or all BALF samples. In terms of overlap between BALF and serum, the IgGome and Fc-glycans are superior with approximately >80% of all features detected in both sample types (compared to <50% for other features).

**Figure 3 ijms-20-02157-f003:**
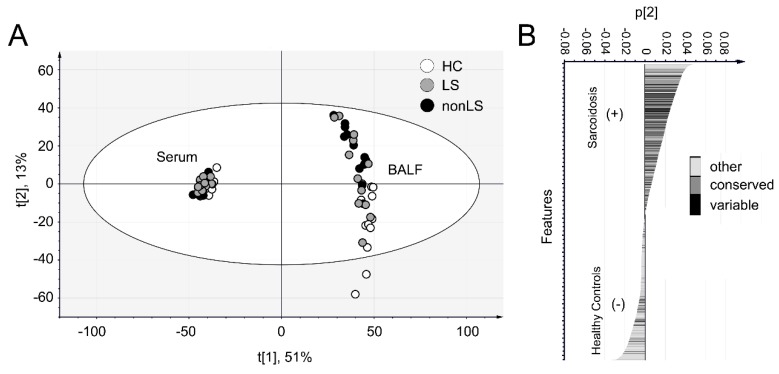
PCA of the complete data set including all features. (**A**) Scores plot. The profiles of the serum and BALF samples are distinctly different as observed by distinct separation along component 1 (*x*-axis). The controls and sarcoidosis patients (LS and nonLS) BALF samples differentiate along component 2 (*y*-axis), indicating that their profiles are different according to disease. (**B**) Loading plot of component 2, i.e., the component that is disease correlating in the BALF. Note that many features that correlate with sarcoidosis are from the conserved and variable regions of IgG.

**Figure 4 ijms-20-02157-f004:**
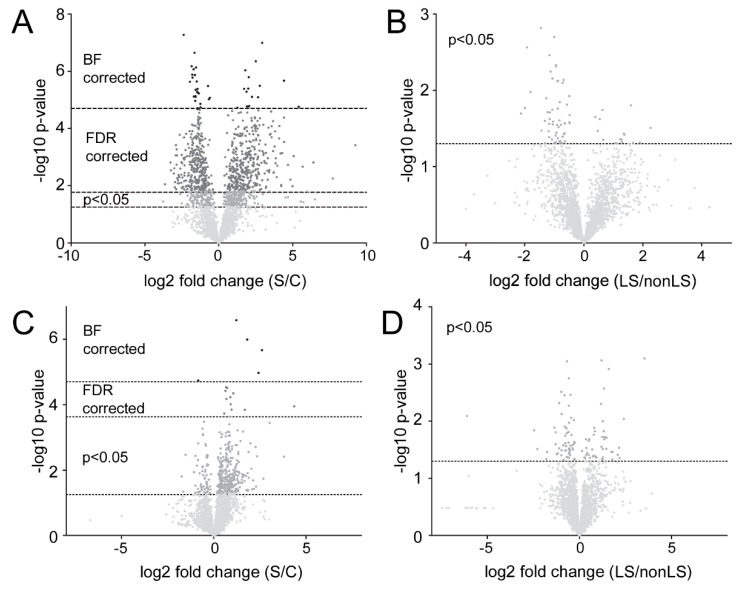
Volcano plots of the data. (**A**) Volcano plot of BALF-controls vs. BALF-sarcoidosis. Negative fold change signifies control correlation, positive fold change indicates sarcoidosis correlation. (**B**) Volcano plot of BALF-LS vs. BALF-nonLS. Negative fold change signifies nonLS correlation, positive fold change indicates LS correlation. (**C**) Volcano plot of the serum-controls vs. serum-sarcoidosis. Negative fold change signifies control correlation, positive fold change indicates sarcoidosis correlation. (**D**) Volcano plot of the serum-LS vs. serum-nonLS. Negative fold change signifies non-LS correlation, positive fold change indicates LS correlation. Significance is indicated by Bonferroni (BF) correction, FDR correction, and *p* < 0.05. The BF correction was adjusted to the total number of variables (*n* = 1711) giving a cut off of 4.71. The FDR was corrected according to the p-value distribution for each data set separately. The big difference in FDR correction between Figure 4A, (cut off = 1.77) and Figure 4C (cut off = 3.63) is due to the much larger portion of variables in the BALF that were significantly different (*p* < 0.05) between controls and patients compared to what was observed in the serum.

**Figure 5 ijms-20-02157-f005:**
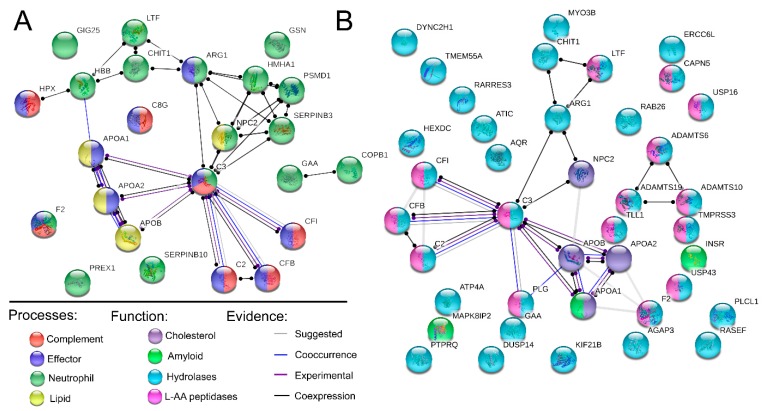
STRING pathway analysis. Analysis of non-IgG proteins and peptides found in significantly elevated levels in BALF of sarcoidosis patients following FDR correction. (**A**) Significantly elevated biological processes. (**B**) Significantly elevated proteins of a specific molecular function. Network links are shown according to evidence mode. For more details see Table 1. AA; amino acid. The gene name abbreviations are given in the Abbreviations.

**Figure 6 ijms-20-02157-f006:**
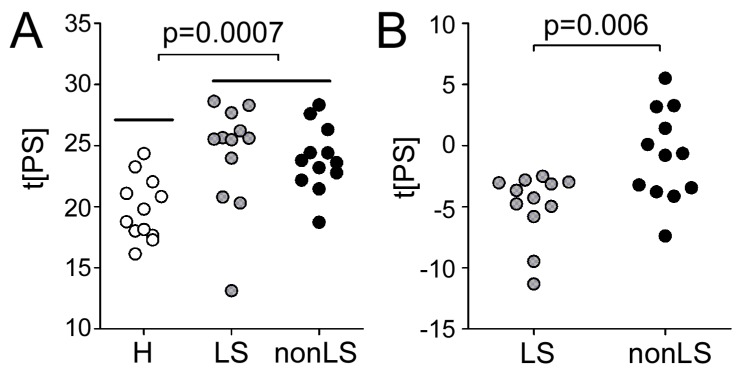
Predictive scores of the serum profiles. Predictive scores of the serum profiles using (**A**) information obtained from the OPLS-DA models of BALF-controls vs. BALF-sarcoidosis and (**B**) information obtained from BALF-LS vs. BALF-nonLS phenotypes.

**Table 1 ijms-20-02157-t001:** STRING analysis of proteins/peptides that were significantly different in BALF between controls and sarcoidosis patients following FDR-correction. Note that many of the proteins are intertwined and found under several processes, functions, and types. Also, note that the peptidase activities in the control and sarcoidosis groups are different. The gene name abbreviations are given in the Abbreviations.

Type	Correlation	Type	Process/Function	FDR	Gene Name
**Process**	**Sarcoidosis**	**Complement**	regulation of complement activation	2.0 × 10^−3^	C2, C3, C8G, CFB, CFI, F2
			regulation of humoral immune response	2.0 × 10^−3^	C2, C3, C8G, CFB, CFI, F2, HPX
			complement activation	7.2 × 10^−3^	C2, C3, C8G, CFB, CFI
			complement activation, classical pathway	1.7 × 10^−2^	C2, C3, C8G, CFI
			complement activation, alternative pathway	1.9 × 10^−2^	C3, C8G, CFB
		**Effector**	regulation of immune effector process	4.0 × 10^−2^	APOA1, APOA2, ARG1, C2, C3, C8G, CFB, CFI, F2, HPX
		**Neutrophil**	neutrophil activation	2.4 × 10^−3^	ARG1, C3, CHIT1, COPB1, GAA, GIG25, GSN, HBB, HMHA1, LTF, NPC2, PREX1, PSMD1, SERPINB10, SERPINB3
			neutrophil mediated immunity	2.4 × 10^−3^	ARG1, C3, CHIT1, COPB1, F2, GAA, GIG25, GSN, HBB, HMHA1, LTF, NPC2, PSMD1, SERPINB10, SERPINB3
		**Lipid**	cholesterol efflux	7.2 × 10^−3^	APOA1, APOA2, APOB, NPC2
			chylomicron remodeling	7.6 × 10^−3^	APOA1, APOA2, APOB
			chylomicron assembly	1.2 × 10^−2^	APOA1, APOA2, APOB
			plasma lipoprotein particle clearance	2.6 × 10^−2^	APOA1, APOA2, APOB, NPC2
			negative regulation of very-low-density lipoprotein particle remodeling	3.8 × 10^−2^	APOA1, APOA2
	**Controls**	**Inhibitors**	negative regulation of catalytic activity	3.7 × 10^−8^	ANXA2, APC, CD109, CST6, CSTA, CSTB, F11R, FABP4, FETUB, FKBP1A, GAPDH, GCHFR, GSTP1, HSPB1, MICAL1, PARK7, PEBP1, RGS2, SCGB1A1, SERPINB12, SLPI, SPINT2, WFDC2, WFIKKN2
			negative regulation of hydrolase activity	5.3 × 10^−8^	CD109, CST6, CSTA, CSTB, FETUB, GAPDH, MICAL1, PARK7, PEBP1, SERPINB12, SLPI, SPINT2, WFDC2, WFIKKN2
			negative regulation of endopeptidase activity	7.8 × 10^−8^	CD109, CST6, CSTA, CSTB, FETUB, GAPDH, MICAL1, PARK7, PEBP1, SERPINB12, SLPI, SPINT2, WFDC2, WFIKKN2
			negative regulation of proteolysis	7.6 × 10^−7^	CD109, CD55, CST6, CSTA, CSTB, FETUB, GAPDH, MICAL1, PARK7, PEBP1, SERPINB12, SLPI, SPINT2, WFDC2, WFIKKN2
**Function**	**Sarcoidosis**	**Peptidases**	endopeptidase activity	2.1 × 10^−3^	ADAMTS10, ADAMTS19, ADAMTS6, C2, C3, CAPN5, CFB, CFI, F2, LTF, PLG, TLL1, TMPRSS3, USP16
			peptidase activity, acting on L-amino acid peptides	5.8 × 10^−3^	ADAMTS10, ADAMTS19, ADAMTS6, C2, C3, CAPN5, CFB, CFI, F2, LTF, PLG, TLL1, TMPRSS3, USP16, USP43
			hydrolase activity	2.6 × 10^−2^	ADAMTS10, ADAMTS19, ADAMTS6, AGAP3, AQR, ARG1, ATIC, ATP4A, C2, C3, CAPN5, CFB, CFI, CHIT1, DUSP14, DYNC2H1, ERCC6L, F2, GAA, HEXDC, KIF21B, LTF, MYO3B, PLCL1, PLG, PTPRQ, RAB26, RARRES3, RASEF, TLL1, TMEM55A, TMPRSS3, USP16, USP43
			serine-type endopeptidase activity	4.3 × 10^−3^	C2, C3, CFB, CFI, F2, LTF, PLG, TLL1, TMPRSS3
		**Cholesterol**	cholesterol transporter activity	4.3 × 10^−3^	APOA1, APOA2, APOB, NPC2
			cholesterol binding	2.6 × 10^−2^	APOA1, APOA2, APOB, NPC2
		**Lipid**	high-density lipoprotein particle receptor binding	2.6 × 10^−2^	APOA1, APOA2
			apolipoprotein receptor binding	4.1 × 10^−2^	APOA1, APOA2
			lipoprotein particle receptor binding	4.1 × 10^−2^	APOA1, APOA2, APOB
			phosphatidylcholine-sterol O-acyltransferase activator activity	4.1 × 10^−2^	APOA1, APOA2
		**Amyloid**	amyloid-beta binding	3.7 × 10^−2^	APOA1, CLSTN1, INSR, MAPK8IP2
	**Controls**	**Inhibitors**	endopeptidase inhibitor activity	6.6 × 10^−7^	CD109, CST6, CSTA, CSTB, FETUB, GAPDH, PEBP1, SERPINB12, SLPI, SPINT2, WFDC2, WFIKKN2
			enzyme inhibitor activity	6.6 × 10^−7^	ANXA2, CD109, CST6, CSTA, CSTB, FETUB, GAPDH, GCHFR, HSPB1, PEBP1, SCGB1A1, SERPINB12, SLPI, SPINT2, WFDC2, WFIKKN2
			peptidase regulator activity	6.6 × 10^−7^	CD109, CST6, CSTA, CSTB, CTSH, FETUB, GAPDH, PEBP1, SERPINB12, SLPI, SPINT2, WFDC2, WFIKKN2
			serine-type endopeptidase inhibitor activity	2.1 × 10^−4^	CD109, PEBP1, SERPINB12, SLPI, SPINT2, WFDC2, WFIKKN2
			cysteine-type endopeptidase inhibitor activity	1.8 × 10^−3^	CST6, CSTA, CSTB, FETUB, WFDC2
			aspartic-type endopeptidase inhibitor activity	1.3 × 10^−2^	GAPDH, WFDC2
			phospholipase A2 inhibitor activity	1.7 × 10^−2^	ANXA2, SCGB1A1
			protease binding	3.3 × 10^−2^	ANXA2, CST6, CSTA, CSTB, PARK7
		**Antioxidant**	antioxidant activity	8.4 × 10^−4^	ALB, CAT, GSTP1, PARK7, SOD1, TXN
		**Peptidases**	cysteine-type endopeptidase activity	3.6 × 10^−2^	CASP14, CTSH, CTSS, TINAG
			peptidase activity	3.6 × 10^−2^	CASP14, CTSH, CTSS, DCD, KLKB1, MYRF, NAPSA, PARK7, PIP, PLG, TINAG
			endopeptidase activity	4.7 × 10^−2^	CASP14, CTSH, CTSS, KLKB1, NAPSA, PIP, PLG, TINAG

**Table 2 ijms-20-02157-t002:** Patient characteristics and available clinical data. Note that compared to the SpotLight data which indicated an elevation of neutrophil-mediated proteins in sarcoidosis, the relative distribution (%) of neutrophils is lower in the patients. This can be explained by a greater increase of the other cell types which would result in a relative decrease in neutrophils (even though the absolute levels are increasing).

	H ^a^	LS ^b^	Non-LS ^c^
*Age (year)*	27 ± 3	42 ± 7	43 ± 9
*Smoking (never/ex)*	12/0	8/3	9/3
*Gender (male/female)*	12/0	12/0	12/0
*X-ray stage I/II/III/IV* ^d^	NM ^e^	5/6/0/0	3/9/0/0
*FVC* ^f^ *% of predicteg* ^f^	114 ± 11	84 ± 16	90 ± 14
*FEV(1)* ^hg^ *% of predicted*	107 ± 9	86 ± 16	86 ± 15
*FEV(1)/FVC* ^i^	79 ± 7	78 ± 7	70 ± 6
*Macrophages % **	86 ± 9	72 ± 21	69 ± 15
*Lymphocytes % **	9 ± 5	26 ± 21	28 ± 16
*Neutrophils % **	2 ± 4	2 ± 1	1 ± 1
*Eosinophils % **	0	0	2 ± 4
*CD4/CD8*	NM	9 ± 7	6 ± 6
*Va2.3^j^%*	NM	18 ± 15	8 ± 8
*CRP* ^k^ *mg/L*	NM	16 ± 26	3 ± 2
*ACE* ^l^ *U/mL*	NM	72 ± 35	64 ± 34
*Alb* ^m^ *g/L*	42 ± 3	41 ± 4	41 ± 4

^a^: Healthy, ^b^: Löfgren Syndrome, ^c^: non-Löfgren Syndrome, ^d^: Stage I indicates granuloma formation in the hilar lymph nodes. Stage II has, in addition to the hilar lymphadenopathy, also granuloma formation in the lung shown as diffuse infiltrates on the x-ray. Stage III has parenchymal infiltrates on the x-ray but an absence of hilar adenopathy, stage IV indicates irreversible pulmonary scarring. ^e^: not measured, ^f^: Forced vital capacity, ^g^: LS group is missing one value. ^h^: Forced expiratory volume 1 sec, ^i^: LS is missing one value, the non-LS group is missing three values. ^j^: V alpha 2.3 T-cell receptor positive, ^k^: C reactive protein, ^l^: Angiotensin-converting enzyme, ^m^: Serum albumin, * Data was obtained from the BAL.

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
