# Peer review of "SpotLight Proteomics—A IgG-Enrichment Phenotype Profiling Approach with Clinical Implications"

_ijms, 2019, doi:10.3390/ijms20092157_

Round 1
Reviewer 1 Report
The paper entitled "SpotLight proteomics – an IgG-enrichment phenotype profiling approach with clinical implications" is interesting in respect to the topic. The paper is well written, and has merit of publication.
Author Response
We thank the reviewer for their positive feedback.
Reviewer 2 Report
The abstract can be improved to show a snapshot of the results.
PCA plot of three serum groups could be included.
Figure 4: y axis is the -log10pvalue, why the FDR and BF corrected regions are different from one plot to another.
Author Response
The abstract can be improved to show a snapshot of the results.
- We agree with the reviewer that the abstract can be improved. Thus, we have changed the abstract to give more focus on the most prominent results.
PCA plot of three serum groups could be included.
- We are now providing an additional figure in the supplemental material (Supplemental Figure 1), which is showing the PCA analysis of the serum and BALF data separately. The Figure is referred to on page 3, line 91-92.
Figure 4: y axis is the -log10pvalue, why the FDR and BF corrected regions are different from one plot to another.
- We thank the reviewer for pointing this out so that we can clarify. The BF correction was adjusted to the total number of variables (n=1711) giving a cut off of 4.71. The FDR was corrected according to the p-value distribution for the BALF data set (cut off=1.77) and the p-value distribution for the serum data set (cut off=3.63) separately. We are now clarifying this in the Figure 4 text section. The big difference in FDR correction between the two data sets is due to the much larger portion of variables in the BALF with a p<0.05 compared to what was observed in the serum.
Reviewer 3 Report
The Manuscript ijms-467593, entitled “SpotLight proteomics – an IgG-enrichment phenotype profiling approach with clinical implications” is an interesting work focalizing on phenotypic stratification in sarcoidosis patients for a precision medicine purpose. In particular, they applied the SpotLight approach in order to highlight a hidden IgG proteome in BAL and serum. They extrapolate new phenotype information in BAL that are reflected in serum, leading to believe that in a future could be possible to stratify patients, from a phenotypic point of view, also starting from an easier recoverable sample like the serum. The paper is clear and well written. Results are interesting and clearly reported.
Minor revision:
The ethical committee review board number is missing.
Author Response
The ethical committee review board number is missing.
- We thank the reviewer for pointing this out. The number is now included in the text on page 11, line 259.